# Multifunctional Characteristics of BCTH:0.5% Sm^3+^ Ceramics Prepared via Hydrothermal Method and Powder Injection Molding

**DOI:** 10.3390/ma16216910

**Published:** 2023-10-27

**Authors:** Rong Zhuang, Qiushi Wang, Bijun Fang, Shuai Zhang, Xiaolong Lu, Jianning Ding

**Affiliations:** 1School of Materials Science and Engineering, Jiangsu Collaborative Innovation Center of Photovoltaic Science and Engineering, Jiangsu Province Cultivation Base for State Key Laboratory of Photovoltaic Science and Technology, National Experimental Demonstration Center for Materials Science and Engineering, Changzhou University, Changzhou 213164, China; zr_youmu@163.com (R.Z.); s21010856057@smail.cczu.edu.cn (Q.W.); shuaizhang@cczu.edu.cn (S.Z.); xll@cczu.edu.cn (X.L.); 2School of Mechanical Engineering, Yangzhou University, Yangzhou 225127, China

**Keywords:** BCTH, Sm^3+^ doping, hydrothermal method, photoluminescence performance, multifunctional performance

## Abstract

Briefly, 0.005-mol Sm^3+^-doped (Ba_0.85_Ca_0.15_)(Ti_0.9_Hf_0.1_)O_3_ ([(Ba_0.85_Ca_0.15_)_0.995_Sm_0.005_](Ti_0.9_Hf_0.1_)O_3_, BCTH:0.005Sm^3+^) lead-free ceramics were prepared via hydrothermal method and powder injection molding using paraffin and oleic acid as binders, and the effects of preparation method and sintering conditions on microstructure, dielectric behavior and optical properties were investigated. XRD Rietveld refinement reveals the coexistence of orthogonal, rhombohedral and tetragonal phases, in which the crystal structure and phase fraction are influenced greatly by sintering temperature and holding time. The ceramics present enhanced relaxor behavior and frequency dispersion phenomenon as compared with those prepared by the solid-state sintering method, and the diffusive index γ value is within 1.421–1.673. The transition mechanism and luminescence performance of BCTH:0.005 Sm^3+^ were analyzed by Blasse formula, photoluminescence spectrum and fluorescence lifetimes, where emission peaks show slight blueshift, fluorescence decay lifetime becomes shorter, electric multipole interaction dominates the energy transfer mechanism, and the down-conversion luminescence is one-photon absorption process. The CIE chromaticity color coordinate (0.4746, 0.5048), correlated color temperature 3134 K and color purity 93.58% are achieved, which reveals that the BCTH:0.005 Sm^3+^ ceramics express high quality yellow emission rather than orange-red light of the hydrothermal method synthesized nano-powder, and have potential application in optical field.

## 1. Introduction

Piezoelectric ceramic is a kind of functional material which can realize the transformation between electrical and mechanical properties. The most widely used piezoelectric ceramics are lead-based (Pb(Zr_1−x_Ti_x_)O_3_, PZT) ceramics [1]. Since lead causes great harm to environmental protection and human health, lead-free piezoelectric ceramics have attracted extensive attention in scientific researchers [2,3,4]. Therefore, developing high piezoelectric performance novel lead-free piezoelectric ceramics to replace the PZT-based piezoelectric ceramics has become the goal of the broad scientific research teams [5,6]. At present, the most widely used piezoelectric ceramics have perovskite structure, which are widely used in thermal, electrical, optical and magnetic fields, and play an important role in industry, agriculture, manufacturing and medical fields [7,8].

Fluorescent piezoelectric ceramic is a new type of multifunctional material, which converts electrical energy, optical energy and mechanical energy into each other by excited fluorescence [9]. Most fluorescent piezoelectric ceramics are formed by doping transition metal ions or rare-earth ions as luminescent centers into the piezoelectric ceramic matrix [10]. Due to product effect between piezoelectricity of ferroelectrics and photoluminescence excited by rare-earth fluorescent center, novel multifunctional application can be generated.

Rare-earth has extensive application in electronic components, laser and magnetic materials [11]. Rare-earth elements are relatively abundant in the earth’s crust and are perfect dopants for studying fluorescent piezoelectric ceramics [12]. The trivalent rare-earth activator ions show pathognomonic emission spectrum through the f-f electron transition, which occurs between the f orbital that are not sensitive to the rear-earth ion environment, and is masked by the electrons of the 5 s and 5p orbital [13]. The rare earth Sm^3+^ possesses 1994 energy levels, whose electrons absorb energy and transmit between different energy levels in the 4f orbital of Sm^3+^ under the excitation of light, and mainly emit orange-red light, being an important luminescent material activator [14,15]. It is reported that rare-earth doped piezoelectric ceramic system presents enhanced piezoelectric ferroelectric and photoluminescence properties [16,17,18]. For example, Wang et al. observed mechanical-electrical, mechanical-optical and electrical-optical coupling in Pr-doped Ba_0.77_Ca_0.23_TiO_3_ ceramics [9]. Liu et al. found that high concentration Sm^3+^ doped BaTiO_3_ ceramics could exacerbated structural inhomogeneity, and new dielectric permittivity plateaus and relaxation peaks appeared in the intermediate frequency range [19]. Such phenomena indicate that rare-earth element doping not only improves electrical properties and induces fluorescence performance, but also produces coupling effect, which has important research significance for developing new multifunctional materials.

BaTiO_3_ ceramics are typically perovskite ferroelectric with high dielectric permittivity and are very important dielectric materials. However, the piezoelectric properties of pure BaTiO_3_-based ceramics are still inferior to that of the PZT-based piezoelectric ceramics. Constructing morphotropic phase boundary (MPB) is a key factor to improve piezoelectric properties, where enhanced electrical properties are induced due to almost no barrier and easy polarization rotation in the polymorphic phase region [20,21]. Therefore, many scholars have introduced one or more components into the BaTiO_3_ matrix to seek novel MPB of multi-component systems to improve electrical properties [22,23,24]. In 2009, Liu et al. prepared 0.5Ba(Ti_0.8_Zr_0.2_)O_3–_0.5(Ba_0.7_Ca_0.3_)TiO_3_ ceramics with piezoelectric coefficient d_33_~620 pC/N and curie temperature T_C_~90 ℃ by traditional solid-state reaction method, and predicted that the d_33_ value in single crystal form can reach 1000~1500 pC/N [25]. In 2016, Zhao et al. prepared (Ba_1−y_Ca_y_)(Ti_1−x_Hf_x_)O_3_ ceramics by traditional solid-state reaction method, and found that the phase boundary is highly dependent on the concentration of Ca^2+^ and Hf^4+^, and the modified ceramics can make the characteristic peak of BaTiO_3_ move and eventually form two-phase or multi-phase coexistence regions [26]. (Ba_1−x_Ca_x_)(Ti_0.95_Hf_0.05_)O_3_ (BCTH) ceramics prepared by Wang et al. reported in 2018 also have good electrical properties near the MPB region [27]. Therefore, according to the existing research results, BCTH ceramics have high electrical properties and development potential.

Since the physical and chemical properties of materials depend partly on composition and partly on structural crystal phase, grain shape and size, the preparation of highly reactive powder is important for modern ceramic manufacturing [28,29]. Hydrothermal synthesis method prepares nano-powders in aqueous solutions under high pressure and temperature conditions. Such process occurs in solution, providing a stable and uniform environment for the formation of materials, and the high temperature and high-pressure conditions during synthesis help minimize defects and promote the formation of high-quality materials [30]. He et al. prepared Nd-doped Na_0.5_Bi_0.495_Nd_0.005_TiO_3_ (NBT-Nd) nano-powder by hydrothermal method, where the obtained ideal results had good crystallization at relatively low reaction temperature and alkali concentration, and the prepared NBT-Nd powder basically did not contain impurity [31]. Lu et al. prepared Ba_0.85_Ca_0.15_Zr_0.1_Ti_0.9_O_3_ (BCZT) ceramic system similar to the BCTH system by hydrothermal method, and the sintering temperature for the BCZT ceramics prepared by hydrothermal method was reduced by 200 °C compared with the solid-phase reaction method [32]. These research results show that the hydrothermal synthesis method not only greatly reduces the temperature of synthesizing compounds, but also has high activity and purity of the prepared powder.

In this paper, 0.5 mol% Sm^3+^-doped [(Ba_0.85_Ca_0.15_)_0.995_Sm_0.005_](Ti_0.9_Hf_0.1_)O_3_ (abbreviated as BCTH:0.005Sm^3+^) ceramics were prepared via hydrothermal method and powder injection molding using paraffin and oleic acid as binders. (Ba_0.85_Ca_0.15_)(Ti_0.9_Hf_0.1_)O_3_ was chosen as phosphor matrix since such composition approaches MPB region [23,26], and 0.5 mol% Sm^3+^ doping amount was selected based on preliminary research [33]. The effects of preparation process, sintering conditions and grain size on the crystal structure, microstructure and electrical and optical properties of the BCTH:0.005Sm^3+^ ceramics were investigated. The mechanism of Sm^3+^ doping induced photoluminescence was discussed, and BCTH:0.005Sm^3+^ has prospect application in optical field due to rare-earth element Sm^3+^ doping.

## 2. Experimental Details

Briefly, 0.5 mol% Sm^3+^-doped [(Ba_0.85_Ca_0.15_)_0.995_Sm_0.005_](Ti_0.9_Hf_0.1_)O_3_ (abbreviated as BCTH:0.005Sm^3+^) ceramics were prepared via hydrothermal method and powder injection molding using paraffin and oleic acid as binders. High purity raw materials BaCl_2_·2H_2_O, CaCl_2_, HfO_2_, TiCl_4_ and Sm_2_O_3_ were weighted according to the designed stoichiometric chemical ratio. After evenly mixing, the weighted raw materials were poured into the polytetrafluoroethylene lining of the hydrothermal reactor. The alkali concentration NaOH was tailored to 14 mol·L^−1^ and the lining filling rate was 75% using distilled water. The hydrothermal reaction was carried out at 180 °C for 24 h, and then naturally cooled to room temperature. The resulting suspension was washed several times to neutrality by centrifugation and separation, and then dried at 80 °C for 10 h to obtain BCTH:0.005Sm^3+^ powder.

Powders obtained by hydrothermal method were used to prepare ceramics by powder injection molding using paraffin wax and oleic acid as injection binder. The well mixed powder was injected into round pellets with 16 mm in diameter and 1~2 mm in thickness under uniaxial pressure of 400 MPa and 65 °C for 10 min. After a very slow degrease process using 1250 °C calcined Al_2_O_3_ powder as adsorbent, the binder-removed pellets were sintered at 1280~1320 °C for 4~8 h, respectively.

The phase structure of the BCTH:0.005Sm^3+^ ceramics prepared under different sintering conditions were characterized by X-ray diffractometer (XRD, Rigaku D/max-2500/PC, Rigaku Corp., Tokyo, Japan) with Cu-Kα1 radiation. Micro-morphology of ceramics’ free surface was observed by the JSM-IT100 scanning electron microscope (SEM, JEOL Ltd., Tokyo, Japan). The photoluminescence performance measurement was undertaken by a spectrophotometer using Xenon Light laser (FlS1000, Edinburgh Instruments, Edinburgh, UK). For dielectric performance characterization, a silver electrode was formed by firing silver paste at 650 °C for 30 min. The dielectric permittivity–temperature curves were measured by Partulab HDMS-1000 (Wuhan Partulab Technology Co., Ltd., Wuhan, China) connected with Microtest 6630–10 (Microtest Corporation, Taipei City, Taiwan) from room temperature to 180 °C.

## 3. Results and Discussion

### 3.1. Phase Structure and Morphology

Figure 1a–e show the XRD patterns of the BCTH:0.005Sm^3+^ ceramics prepared by hydrothermal methods measured in the 2θ range of 10–80° at room temperature. To further judge phase evolution, the amplified XRD patterns with 2θ = 44–46° are also presented. The standard diffraction peaks cited from BaTiO_3_ with orthorhombic (O phase), rhombohedral (R phase) and tetragonal (T phase)-symmetries (PDF#81-2200, PDF#85-1796 and PDF#89-1428, respectively, are indicated in Figure 1e with vertical lines for comparison [34]. It is seen that the samples prepared by the hydrothermal method (sintering temperature is 1280 °C to 1320 °C) have a pure phase of perovskite structure, proving the high sintering activity of the nano-sized powder obtained by the hydrothermal method where slight impurity phase exists [34]. As comparison with higher sintering temperature of normally 1450 °C and above used in the solid-state method, the decreased sintering temperature in this work reveals the high sintering activity of the nano-powder obtained by hydrothermal method [35]. The XRD patterns do not change obviously with increasing sintering temperature and holding time, except slight widening of diffraction peaks and tendency of moving to high 2θ angle of diffraction peak position as shown in the expanded XRD patterns for the hydrothermal method prepared ceramics (Figure 1a–d). According to the peak characteristic of R phase, the broadening of a single peak suggests the increase in R phase amount in the multi-phase coexistence structure, and the high angle movement of diffraction peaks shows shrinkage of lattice cell. Compared with pure BCTH ceramics prepared by Wang et al. by solid sintering method also using powder injection molding, the samples prepared by hydrothermal method have similar peak shape, and present single and uniform diffraction peak at 2θ = 44–46° [35]. In general, all sintered BCTH:0.005Sm^3+^ ceramics exhibit a standard perovskite structure, indicating that the doped cations have been successfully dissolved into the BaTiO_3_ matrix.

For the purpose of obtaining deeper insight into the phase structure of the studied samples, the detailed crystal structure information of BCTH:0.005Sm^3+^ was investigated by Rietveld refinement (2θ = 10–80°) with GSAS. Figure 2a,b shows the refined XRD patterns of the ceramics prepared by hydrothermal methods sintered at 1300 ℃ for 8 h and 12 h and the inset show the single (111) peak fitting plot. The XRD data of reference standard materials used for Rietveld refinement in Figure 2a,b are shown in supporting information. As presented, all calculated and measured diffraction peaks are consistent well with each other. The calculated lattice parameters and Rietveld refinement parameters are shown in Table 1. In the case of the same holding time, the O-R phase boundary exists in ceramics sintered at 1280 °C, and with the increase in sintering temperature, O, R and T phases co-exist in ceramics. When the sintering temperature reaches 1320 °C, the ceramics become co-existence of O-R phases again. Therefore, sintering temperature affects phase structure of the ceramics. When maintaining sintering temperature at 1300 °C and increasing holding time, it is found that the ceramics change from O-R-T three phases coexistence to only O-R two phases, indicating that holding time also changes phase structure of the ceramics. The R phase tends to decrease with increasing sintering temperature and increase with prolonging holding time in the ceramics prepared by the hydrothermal method, which indicates that sintering temperature and holding time influence the phase boundary region. The disadvantage of the solid-state method is that the sintering temperature is too high (1450 °C) [35]. By comparison, the hydrothermal method can indeed reduce the sintering temperature for preparing ceramics, and the preparation process of the hydrothermal method can be improved further to enhance structural uniform and tailor physical performance [27].

Figure 2c presents the crystal structure of the BCTH host, where Ba^2+^ (0.161 nm, 12 coordination) occupies eight positions at the apex of the cubic structure, O^2−^ (0.140 nm, 6 coordination) occupies the six face center positions of the cube, and Ti^4+^ (0.061 nm, 6 coordination) occupies the center position of the cube. Since the radius of Ca^2+^ (0.134 nm, 12 coordination) is close to that of Ba^2+^ and the radius of Hf^4+^ (0.071 nm, 6 coordination) is close to that of Ti^4+^, Ca^2+^ occupies the position of Ba^2+^ and Hf^4+^ occupies the position of Ti^4+^. The purple plane in the Figure 2c shows the (111) crystal plane of the BCTH host. The possibility of Sm^3+^ (0.124 nm, 12 coordination) substituting the Ba^2+^ ions can be evaluated by the radii ratio calculated using equation [36]:(1)$$D_{r}=\frac{R_{m}(CN)-R_{d}(CN)}{R_{m}(CN)}$$
where *R_m_*(*CN*) and *R_d_*(*CN*) represent the radius of the host ions and dopant ions, respectively, and *CN* is the co-ordination number. The calculated radius ratio percentage is 22.9%. Since substitution can occur when radius difference percentage around the matrix and dopant ions is less than 30%, the Sm^3+^ ions tend to occupy the Ba^2+^ sites in the lattice.

In order to clearly observe the microscopic morphology and the grains size distribution, Figure 3 shows the SEM images of the BCTH:0.005Sm^3+^ ceramics, and insets show histograms of the grain size distribution of the corresponding samples. It could be clearly found that the grain size of the ceramics prepared by the hydrothermal method is rather small, and the grains gradually grow as the sintering temperature and the holding time increase. The average grain size increases from 0.68 μm to 1.33 μm with a growth rate of 95.6% when the sintering temperature increases from 1280 °C to 1320 °C; however, the average grain size increases from 0.69 μm to 0.89 μm, with a growth rate of only 29% when the holding time increases from 4 h to 12 h. Therefore, the change in holding time has little effect on grain growth as compared with sintering temperature. As comparison, the grain size of pure BCTH ceramics prepared by solid-state method via powder injection molding reported by Wang et al. is 8.38 μm sintered at 1450 ℃ for 4 h [35], and the grain size of 0.5 mol% Eu^3+^ doped BCTH ceramics prepared by solid-state method reported by Zhang et al. is 10.3 μm sintered at 1450 °C for 3 h [37]. The maximum grain size of the BCTH:0.005Sm^3+^ ceramics is 1.33 μm prepared by the hydrothermal method sintered at 1320 °C for 8 h, as recorded in Figure 3(e2), being much smaller than that of ceramics prepared by solid-state method.

The decrease in grain size in this work can be mainly attributed to the decreased sintering temperature by the hydrothermal method. Comparing the work by Wang et al. and Zhang et al., rare-earth doping may promote grain growth during sintering [35,37], which can be ignored in this work as compared with using hydrothermal technique according to the above analysis (insets of Figure 3). All ceramics have densified and polyhedron-shape micro-morphology with clear grain boundary, whereas the hydrothermal method synthesized ceramics present apparent bimodal characteristic of grain size. Such phenomena indicate that solid-state sintering mechanism takes major role in the densification of both methods sintered BCTH:0.005Sm^3+^ ceramics [38,39], and the powder obtained by the hydrothermal method is nano-sized particle which has high sintering active and can be sintered into densified ceramics at lower sintering temperature without excessive grain growth.

### 3.2. Electrical Performance

Figure 4 and Table 2 show the effects of sintering temperature and holding time on the dielectric properties of BCTH:0.005 Sm^3+^ ceramics prepared by hydrothermal method. It can be seen that sintering temperature and holding time have a great influence on the width of dielectric peak, the value of dielectric permittivity maximum (ε_m_), the temperature corresponding to the ε_m_ value (T_C_), and the dielectric frequency dispersion characteristic. The dielectric peak becomes apparently wider accompanied by an irregular change in dielectric permittivity under different sintering conditions. The dielectric loss of all ceramics is at a low level. The rapid decrease in loss tangent around room temperature correlates with the occurrence of low-temperature ferroelectric phase transition. As shown in Figure 4b, T_C_ increases first and then decreases slowly, ε_m_ presents upward trend, and tanδ decreases first and then increases with the increase in sintering temperature under 8 h holding time. Figure 4c shows that with the elongation of holding time under 1300 °C sintering temperature, T_C_ rises gradually, ε_m_ rises first and then falls, and tanδ declines first and then becomes stable. Sintering temperature and holding time have significant effects on the dielectric performance of the BCTH:0.005 Sm^3+^ ceramics prepared by hydrothermal method, where the overall performance of the BCTH:0.005 Sm^3+^ ceramics sintered at 1300 °C for 8 h is the best.

In general, ceramics sintered at 1300 °C for 8 h have the best comprehensive performance. Therefore, Figure 5a shows the dielectric temperature spectra of BCTH:0.005Sm^3+^ ceramics prepared by hydrothermal method at 1300 °C for 8 h at different frequencies. As can be seen from the figure, the Curie temperature of BCTH:0.005Sm^3+^ ceramics prepared by hydrothermal method is 82 °C, which is somewhat lower than that of the Eu-doped BCTH ceramics prepared by Zhang et al. by solid phase method (T_C_ = 86 °C) [37]. BCTH:0.005Sm^3+^ ceramics prepared by hydrothermal method have lower dielectric permittivity, wider dielectric peak and dielectric frequency dispersion. Especially at low frequency, dielectric loss is larger. This can be attributed to the point charge defects caused by the strong volatilities of the nano-powders prepared by hydrothermal method in the sintering process, thus showing the behavior characteristics of thermal activation jump loss [39].

Figure 5b shows the fitting of the exponential law of BCTH:0.005Sm^3+^ ceramics prepared by hydrothermal method sintered at 1300 °C for 8 h, and Table 3 shows the diffusive index γ of BCTH:0.005Sm^3+^ ceramics prepared by hydrothermal method at different sintering conditions. As shown in Table 3, the γ value decreases first and reaches minimum value of 1.456 at 1300 °C, then increases to maximum value of 1.612 at 1310 °C, and, finally, decreases again when increasing sintering temperature with same holding time of 8 h, whereas the γ value increases gradually with increasing holding time at same sintering temperature of 1300 °C. The increase in γ value reveals the enhancement of relaxor phenomenon. The γ value is within 1.421–1.673, showing non-typical relaxor ferroelectrics. Appendix A shows dielectric behavior characteristic fitted by the Curie–Weiss law using BCTH:0.005Sm^3+^ ceramics prepared by hydrothermal method sintered at 1300 °C for 8 h as an example, in which the fitting also presents deviation. Therefore, the BCTH:0.005Sm^3+^ ceramics prepared by hydrothermal method present complex dielectric characteristic, which can be attributed to normal ferroelectrics with apparent dispersion type phenomenon depending on sintering conditions [40,41,42].

### 3.3. Optical Properties

Figure 6 shows the excitation spectra of the BCTH:0.005Sm^3+^ ceramics obtained at the wavelength of 596 nm. Similar excitation spectra are obtained within the sintering temperature range, where the intensity of the excitation peaks tends to decrease upon increasing sintering temperature with slight irregularity. Obvious excitation peaks are observed at 380 nm, 409 nm, 421 nm, 464 nm and 480 nm, corresponding to the energy level transitions of ^6^H_5/2_ → ^6^P_7/2_, ^6^H_5/2_ → ^4^F_7/2_, ^6^H_5/2_ → ^6^P_5/2_, ^6^H_5/2_ → ^4^I_13/2_ and ^6^H_5/2_ → ^4^I_11/2_ of Sm^3+^, respectively. among which, the 409 nm excitation peak is much stronger, which is chosen as excitation light to excite all ceramics.

Figure 7 shows the emission spectra of the BCTH:0.005Sm^3+^ ceramics prepared by hydrothermal method at an excitation wavelength of 409 nm. Characteristic transitions ^4^G_5/2_ → ^6^H_5/2_ at 557 nm and ^4^G_5/2_ → ^6^H_7/2_ at 591 nm relate to both magnetic dipole (MD) and electric dipole (ED) transitions, where ^4^G_5/2_ → ^6^H_5/2_ mainly depends on the MD transition, and ED transition is the main proportion in ^4^G_5/2_ → ^6^H_7/2_. Characteristic transitions ^4^G_5/2_ → ^6^H_9/2_ at 637 nm and ^4^G_5/2_ → ^6^H_11/2_ at 699 nm are both ED transition, and the higher the ED transition intensity is, the more obvious asymmetry exists in the local environmental symmetry around Sm^3+^ ions [14]. The strongest emission peak locates at 591 nm, corresponding to the ^4^G_5/2_ → ^6^H_7/2_ energy level transition of Sm^3+^. Such luminescence intensity change relates to the energy release of Sm^3+^ ions by non-radiative transition in the form of different local lattice vibration, leading to the change in symmetry of the internal environment of Sm^3+^ ions [43]. As compared with the hydrothermal method-synthesized nano-powder, the emission peak shows a slight blue shift for the ^4^G_5/2_ → ^6^H_7/2_ energy level transition, where the same transition occurs at 596 nm for the nano-sized powder [33], mainly correlating with the change in lattice cell of the nano-sized powder and submicron ceramics.

Although sintering conditions do not affect shape and position of the emission spectra overall, the emission peak intensity is affected obviously. The emission luminescence intensity shows a “W” change trend with the increase in sintering temperature, i.e., it decreases first and then tends to increase with slight irregularity, and the 1300 °C sintered ceramics have the strongest emission intensity. By comparison, the luminescence intensity increases to maximum value at 8 h holding time and then deceases with further elongating holding time. Therefore, the phase structure and lattice distortion of ferroelectric matrix affect fluorescence emission intensity.

Generally, the non-radiative energy transfer mechanism can be attributed to radiative reabsorption, exchange interaction, and electrical multistage interaction [43]. Radiation reabsorption is the main way of the non-radiative energy transfer mechanism when the excitation spectrum and emission spectrum overlap more [43]. It can be found from Figure 6 and Figure 7 that the excitation spectra and emission spectra do not overlap, so the energy transfer mechanism of BCTH:0.005Sm^3+^ does not belong to radiation reabsorption. In order to determine the energy transfer mechanism in detail, it is necessary to understand the critical distance of doped ions. The critical transfer distance (*R_c_*) can be calculated by Blasse formula [44,45]:(2)$$R_{c}=2{\left(\frac{3V}{4\pi CN}\right)}^{\frac{1}{3}}$$
where *V* is the volume of the host lattice cell (*V* = 64.56 Å), *N* is the number of lattice points replaced by the dopant in the unit cell (*N* = 8) and *C* is the critical doped concentration (*C* = 0.005). When *R_c_* is less than 5, the exchange interaction is the main mode of energy transfer, otherwise the electric multipole interaction dominates the energy transfer [44,45]. Based on the data shown in above parentheses, the calculated *R_c_* is ~14.55 Å, which is greater than 5, indicating that the electric multipole interaction is the main energy transfer mechanism of the BCTH:0.005Sm^3+^ ceramics.

To figure out the luminescence mechanism, emission spectra of the BCTH:0.005Sm^3+^ ceramics prepared by hydrothermal method sintered at 1300 ℃ for 8 h irradiated by 591 nm laser at various pump power density were obtained, as shown in Figure 8a. It can be seen that the intensity of all emission peaks decreases with the increase in pump power, while the overall shape of emission spectra shows no change except for slight peak broadening. It is well known that in the whole pump power range, the isolated ions show a linear slope of their down-conversion luminescence, and the observed down-conversion luminescence intensity is the sum of the mutual contributions of all ions [46]. The relationship between the luminescence intensity (*I*) and the pump power (*P*) can be described as below [47]:(3)$$I\propto P^{n}$$
where *I* is the emission intensity, *P* presents the excitation pump power and *n* stands for the number of photons which can be calculated by the slope of the fitted line of *ln(I)* vs. *ln(P)*. Figure 8b shows the emission intensity (*I*) corresponding to the ^4^G_5/2_ → ^6^H_5/2_, ^4^G_5/2_ → ^6^H_7/2_, ^4^G_5/2_ → ^6^H_9/2_ and ^4^G_5/2_ → ^6^H_11/2_ transitions of Sm^3+^ as a function of the excitation pump power (*P*). According to the linear fitting results, the *n* value for the down-conversion emission ranges from 1.07 to 1.44, which reveals that the luminescence emission in the BCTH:0.005Sm^3+^ ceramics is dominated by a one-photon assisted down-conversion process [46,48,49].

Figure 9 shows the decay dynamic of the BCTH:0.005Sm^3+^ ceramics. The resulted fluorescence decay lifetime curve can by fitted by the exponential decay function [50]:(4)$$I\left(t\right)=A_{1}exp(-\frac{t}{t_{1}})+A_{2}exp(-\frac{t}{t_{2}})\;$$
where *I(t)* is the luminescent intensity at different time, *A*_1_ and *A*_2_ are the related pre-exponential constant, and *t*_1_ and *t*_2_ are the fitted fluorescent fast lifetime and slow lifetime, respectively. The average decay lifetimes (*t_av_*) can be calculated by converting the above formula:(5)$$t_{av}=\frac{A_{1}t_{1}^{2}+A_{2}t_{2}^{2}}{A_{1}t_{1}+A_{2}t_{2}}$$

The fluorescence decay lifetime is 0.40–0.99 μs of the sintered BCTH:0.005Sm^3+^ ceramics, which is lower than that of the nano-sized powder prepared by hydrothermal method (being 8.98–11.81 μs) [33]. Such phenomenon may be due to the decrease in the distance between the luminescent centers in the ceramics after sintering, which makes the excited electrons more likely to undergo non-radiative relaxation and release energy to return to the ground state, resulting in the decrease in fluorescence lifetime. With the increase in sintering temperature, the fitted fluorescence decay lifetime presents a bimodal increasing-decreasing trend, having maximum lifetime of 0.99 μs when sintered at 1310 °C for 8 h (Figure 9a). When sintered at 1300 °C for different holding times, the fluorescence decay lifetime increases first and then decreases with the increase in holding time, and reaches a maximum lifetime of 0.91 μs at 8 h holding time (Figure 9b). Since crystal micro-morphology has a great influence on the distribution of defects, the fluorescence decay lifetime of the BCTH:0.005Sm^3+^ ceramics prepared under different sintering conditions shows difference [51].

In order to analyze the luminescence mechanism of the BCTH:0.005Sm^3+^ ceramics, the energy level transition diagram of Sm^3+^ in the samples was shown in Figure 10a. When excited by an appropriate excitation light source, the ground state electrons at ground state energy level ^6^H_5/2_ absorb photons to produce transitions and become excited state electrons at different energy levels. Electrons at excited state energy levels are unstable, and tend to radiate surplus energy in form of photons and transmit back to base energy level. The excited electrons of Sm^3+^ in the higher energy lever transit to the lower energy level ^4^G_5/2_ through nonradiative relaxation without photons release, and then electrons at ^4^G_5/2_ level jump back to H-level through radiation relaxation. The H-level splits into several sub-levels and light with wavelength of 561 nm, 596 nm, 643 nm and 703 nm is emitted depending on the energy of the transition from the ^4^G_5/2_ level to the sub-levels [15].

Figure 10b shows the CIE chromaticity color coordinate (0.4746, 0.5048) dominated by the BCTH:0.005Sm^3+^ ceramics sintered at 1300 °C for 8 h excited by 409 nm light, being in the yellow light region rather than in the orange-red light zone of the hydrothermal method synthesized nano-powder [33]. Table 4 lists the chromaticity coordinates of all BCTH:0.005Sm^3+^ ceramics prepared by different sintering conditions. It can be seen that all chromaticity coordinates are close to each other and the emission color of all BCTH:0.005Sm^3+^ ceramics do not change significantly with the change in sintering conditions. The correlated color temperature (*CCT*) can be expressed by the following equation [52,53]:(6)$$CCT=-449n^{3}+3525n^{2}-6823n+5520.33$$
where *n* = (*x* − *x_e_*)/(*y* − *y_e_*), in which (*x*, *y*) is the chromaticity coordinate, and (*x_e_*, *y_e_*) is the coordinate of the epicenter with the value of (0.3320, 0.1858) [52,53]. The results are also lists in Table 4, which shows low color temperature correlation.

The color purity is an important tool to investigate the chromaticity property of phosphors, and luminescent materials with high color purity are often able to adjust colors at a wider level. The color purity of samples is calculated by the following equation [54]:(7)$$Color\;purity=\sqrt{\frac{{\left(x-x_{i}\right)}^{2}+{\left(y-y_{i}\right)}^{2}}{{\left(x_{d}-x_{i}\right)}^{2}+{\left(y_{d}-y_{i}\right)}^{2}}}\times 100\%$$
where (*x*, *y*) is the calculated color coordinate of the sample, (*x_i_*, *y_i_*) is the CIE illuminate coordinate with the value of (0.3333, 0.3333), and (*x_d_*, *y_d_*) is the main emission light coordinate of the emission spectrum. Figure 10b shows that the coordinate position of the sample presents yellow emission light, which is closer to the standard yellow coordinate, so (*x_d_*, *y_d_*) has the value of (0.48, 0.52). The color purity of all BCTH:0.005Sm^3+^ ceramics is calculated and is also shown in Table 4. The color purity changes slightly due to variation in sintering conditions, but all above 93%, showing high color purity and potential application in the field of color development [55].

## 4. Conclusions

BCTH:0.005Sm^3+^ ceramics prepared via hydrothermal method and powder injection molding using paraffin and oleic acid as binders have pure perovskite structure, where Sm^3+^ ions tend to locate at position of Ba^2+^ ions. Based on the XRD Rietveld refinement, sintering conditions have a significant effect on phase structure and phase fraction. Ceramic grains grow gradually with the increase in sintering temperature and holding time, whereas the grain size is much smaller than that of the ceramics prepared by solid-state method due to the high sintering active nano-powder. Due to “nano-size effect”, the BCTH:0.005Sm^3+^ ceramics prepared by hydrothermal method have lower dielectric permittivity, and larger loss tangent, especially at low frequency and approaching to room temperature, relating to low-temperature ferroelectric phase transition. The ceramics exhibit wide dielectric peak, apparent dielectric frequency dispersion, especially at low frequency, and the diffusive index γ value is within 1.421–1.673, indicating that the BCTH:0.005Sm^3+^ ceramics are normal ferroelectrics with apparent dispersion type phenomenon depending on sintering conditions. Compared with the nanosized powder prepared by hydrothermal method, the emission peaks of ceramics show slight blueshift, and the fluorescence decay lifetime becomes shorter, relating to the decrease in distance between the luminescent centers after sintering and the easily undergoing non-radiative relaxation to the ground state. The main energy transfer mechanism of the BCTH:0.005Sm^3+^ ceramics is electric multipole interaction, and the down-conversion luminescence presents one-photon absorption process. The CIE coordinate is (0.4746, 0.5048) in the yellow region, the *CCT* value is 3134 K locating in the warm light region, and the corresponding color purity is 93.58%, which indicate that the BCTH:0.005Sm^3+^ ceramics present potential application in the field of color development.

## Figures and Tables

**Figure 1 materials-16-06910-f001:**
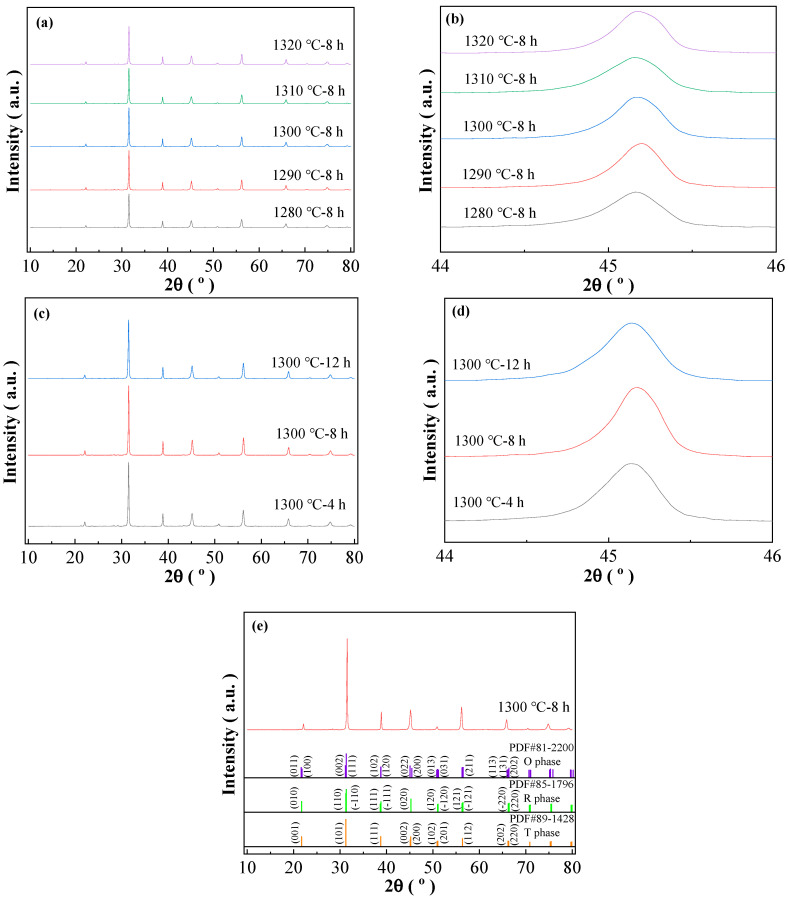
XRD patterns of BCTH:0.005Sm^3+^ ceramics sintering and holding at different temperature and time. (**a**) Samples prepared by hydrothermal method sintering temperature from 1280 °C to 1320 °C. (**c**) Samples prepared by hydrothermal method holding time from 4 h to 12 h. (**e**) Samples prepared by hydrothermal methods sintered at 1300 °C for 8 h with PDF vertical lines for comparison. (**b**,**d**) are their expanding XRD patterns at 2θ = 44–46°.

**Figure 2 materials-16-06910-f002:**
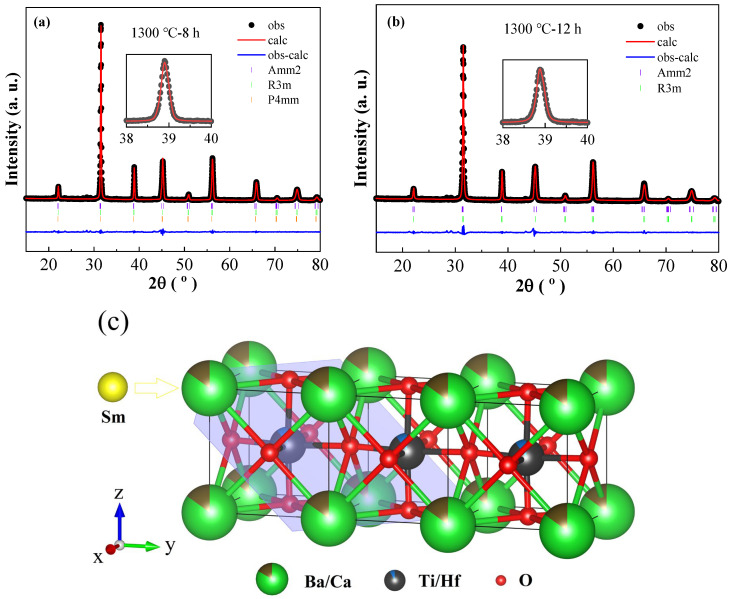
(**a**,**b**) Rietveld refinement of XRD patterns of BCTH:0.005Sm^3+^ ceramics prepared by hydrothermal method sintered at 1300 ℃ for 8 h and 12 h, respectively. (**c**) the crystal structure of BCTH host with Sm^3+^ doping.

**Figure 3 materials-16-06910-f003:**
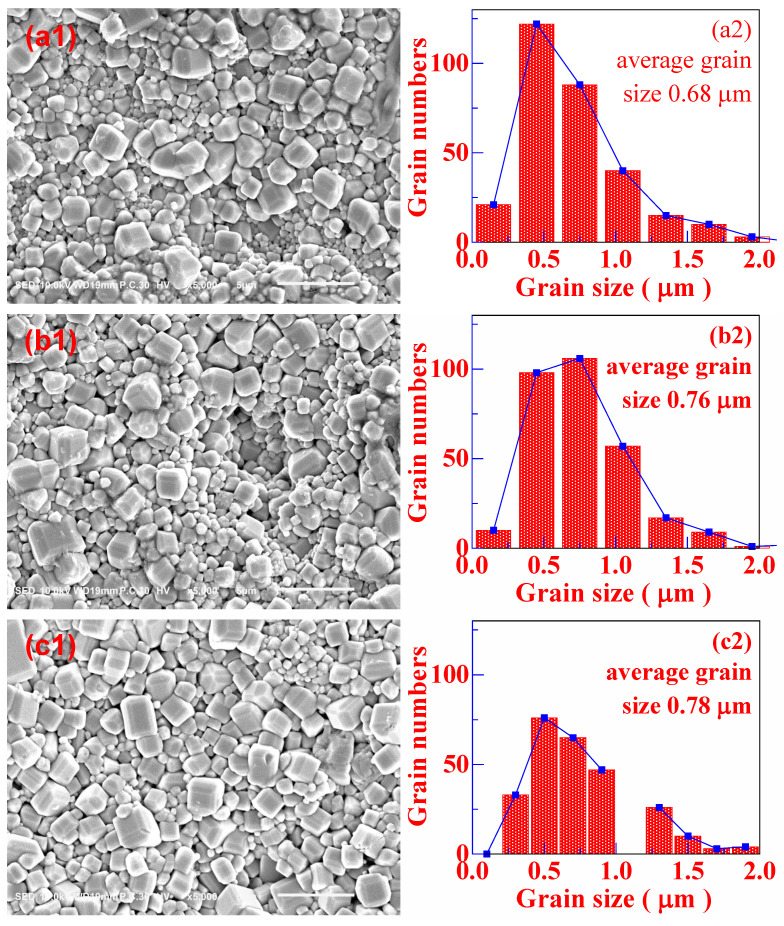
(**1**) SEM image and (**2**) corresponding grain size distribution of BCTH:0.005Sm^3+^ ceramics prepared by hydrothermal method sintered at (**a1**,**a2**) 1280 °C for 8 h; (**b1**,**b2**) 1290 °C for 8 h; (**c1**,**c2**) 1300 °C for 8 h; (**d1**,**d2**) 1310 °C for 8 h; (**e1**,**e2**) 1320 °C for 8 h; (**f1**,**f2**) 1300 °C for 4 h and (**g1**,**g2**) 1300 °C for 12 h.

**Figure 4 materials-16-06910-f004:**
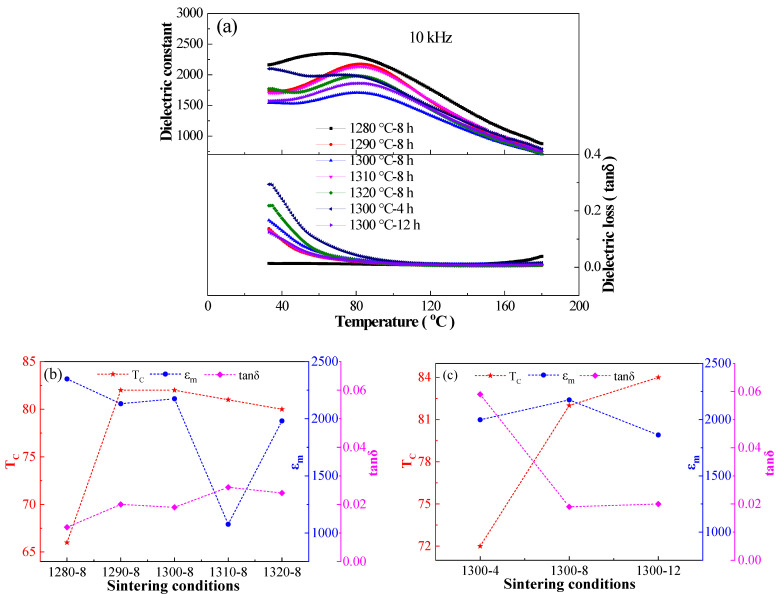
(**a**) Temperature dependence of dielectric permittivity and loss tangent of BCTH:0.005Sm^3+^ ceramics measured at 10 kHz upon heating. (**b**) Sintering temperature dependence of T_C_, ε_m_, and tanδ. (**c**) Holding time dependence of T_C_, ε_m_, and tanδ.

**Figure 5 materials-16-06910-f005:**
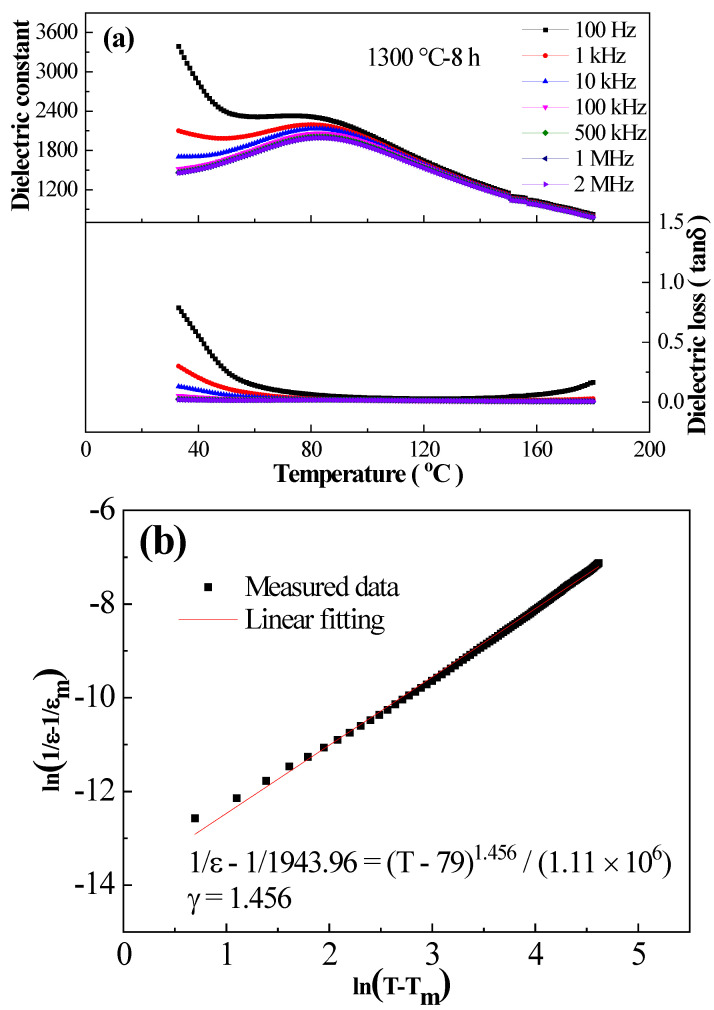
Dielectric performance of BCTH:0.005Sm^3+^ ceramics. (**a**) Prepared by hydrothermal method sintered under 1300 °C for 8 h and (**b**) exponential law fitting using 10 kHz data above the T_m_ temperature.

**Figure 6 materials-16-06910-f006:**
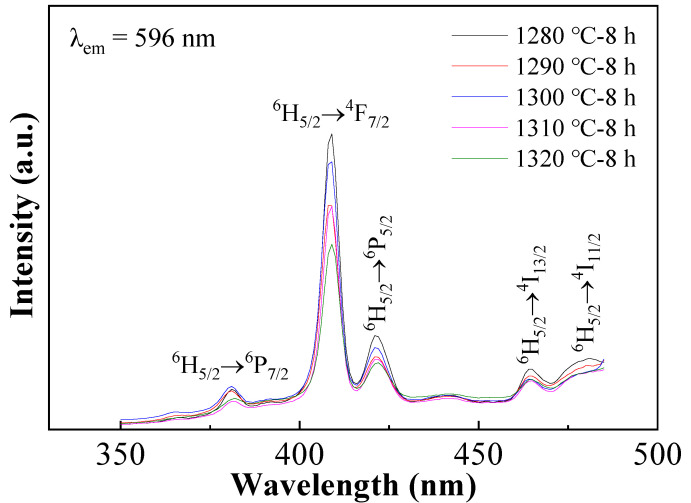
Excitation spectra of BCTH:0.005Sm^3+^ ceramics prepared by hydrothermal method obtained at the wavelength of 596 nm.

**Figure 7 materials-16-06910-f007:**
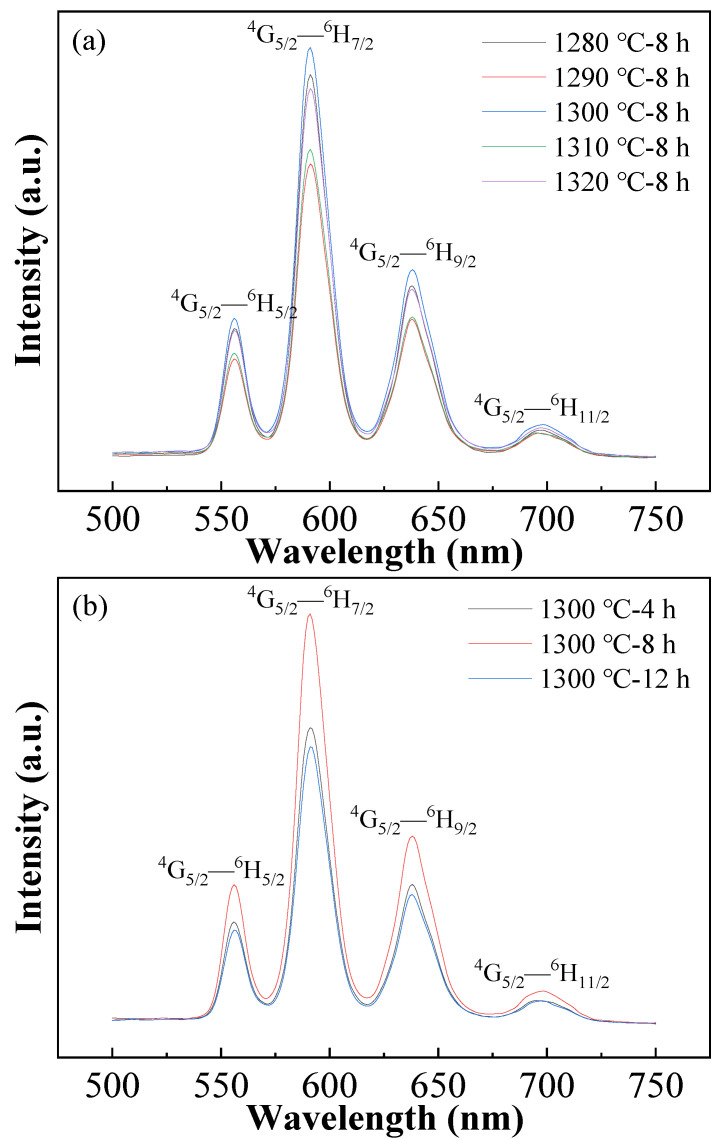
Emission spectra of BCTH:0.005Sm^3+^ ceramics excited by 409 nm light. (**a**) Sintered at different temperature for 8 h; (**b**) sintered at 1300 °C for different holding time.

**Figure 8 materials-16-06910-f008:**
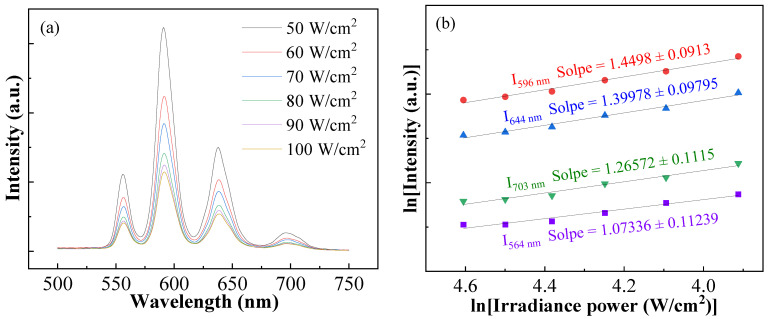
(**a**) Emission spectra of BCTH:0.005Sm^3+^ ceramics prepared by hydrothermal method sintered at 1300 °C for 8 h irradiated by 591 nm laser at various pump power densities. (**b**) Emission intensity (I) corresponding to the ^4^G_5/2_ → ^6^H_5/2_, ^4^G_5/2_ → ^6^H_7/2_, ^4^G_5/2_ → ^6^H_9/2_ and ^4^G_5/2_ → ^6^H_11/2_ transitions of Sm^3+^ as a function of the excitation pump power (P).

**Figure 9 materials-16-06910-f009:**
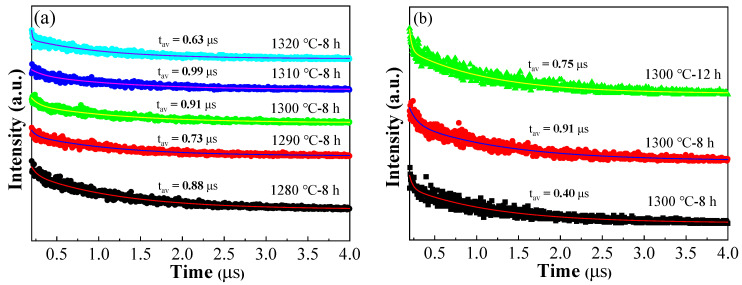
Fluorescence decay lifetime of BCTH:0.005Sm^3+^ ceramics with excitation wavelength of 409 nm and detection wavelength of 596 nm. (**a**) Sintered at different temperature for 8 h holding time; (**b**) sintered at 1300 °C for different holding time.

**Figure 10 materials-16-06910-f010:**
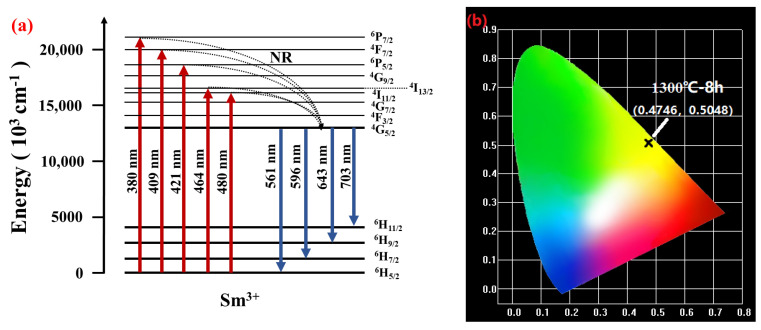
(**a**) Energy level transition diagram of Sm^3+^. (**b**) CIE chromaticity diagram of BCTH:0.005Sm^3+^ ceramics sintered at 1300 ℃ for 8 h under near ultraviolet wavelength of 409 nm excitation.

**Table 1 materials-16-06910-t001:** The lattice parameters and Rietveld-refinement parameters of BCTH:0.005Sm^3+^ ceramics.

Sample	Space Group	a (Å)	b (Å)	c (Å)	α = β = γ (°)	Fraction (%)	Sig	R_w_ (%)
1280 °C-8 h	P4mm					0	2.91	7.60
	Amm2	3.9937	5.6761	5.6871	90	41.73		
	R3m	4.0053	4.0053	4.0053	89.833	58.27		
1290 °C-8 h	P4mm	4.0073	4.0073	4.0443	90	24.44	2.59	6.93
	Amm2	3.9929	5.6781	5.6934	90	31.29		
	R3m	4.0062	4.0062	4.0062	89.873	44.27		
1300 °C-8 h	P4mm	4.0042	4.0042	4.0338	90	22.60	1.97	5.09
	Amm2	3.9939	5.6691	5.6905	90	37.84		
	R3m	4.0053	4.0053	4.0053	90.023	39.56		
1310 °C-8 h	P4mm	4.0114	4.0114	4.0403	90	28.93	2.29	5.91
	Amm2	3.9946	5.6768	5.6883	90	33.87		
	R3m	4.0063	4.0063	4.0063	89.949	37.20		
1320 °C-8 h	P4mm					0	2.32	6.05
	Amm2	3.9940	5.6756	5.6923	90	42.30		
	R3m	4.0068	4.0068	4.0068	89.826	57.50		
1300 °C-4 h	P4mm	4.0117	4.0117	4.0418	90	27.73	2.30	5.95
	Amm2	3.9961	5.6814	5.6935	90	34.40		
	R3m	4.0088	4.0088	4.0088	89.949	37.87		
1300 °C-12 h	P4mm					0	2.43	6.75
	Amm2	3.9916	5.6722	5.6962	90	37.67		
	R3m	4.0043	4.0043	4.0043	89.925	62.33		

**Table 2 materials-16-06910-t002:** ε_m_ and T_C_ of BCTH:0.005Sm^3+^ ceramics measured at 10 kHz upon heating under different sintering conditions.

Sintering Conditions	1280 °C 8 h	1290 °C 8 h	1300 °C 8 h	1310 °C 8 h	1320 °C 8 h	1300 °C 4 h	1300 °C 12 h
ε_m_	2348.7	2131.9	2175.4	1076.5	1981.0	1998.9	1860.4
T_C_ (℃)	66	82	82	81	80	72	84
tanδ	0.012	0.020	0.019	0.026	0.024	0.059	0.020

**Table 3 materials-16-06910-t003:** Diffusive index of BCTH:0.005Sm^3+^ ceramics prepared by hydrothermal method under different sintering conditions.

Sintering Conditions	1280 °C 8 h	1290 °C 8 h	1300 °C 8 h	1310 °C 8 h	1320 °C 8 h	1300 °C 4 h	1300 °C 12 h
Diffusive index	1.517	1.463	1.456	1.612	1.579	1.421	1.673

**Table 4 materials-16-06910-t004:** Lifetime and chromaticity coordinates of BCTH:0.005Sm^3+^ ceramics sintered under different sintering conditions.

Sintering Condition	Lifetime (μs)	CIE (x, y)	*CCT* (K)	Color Purity (%)
1280 °C-8 h	0.88	(0.4750, 0.5040)	3125	93.43
1290 °C-8 h	0.73	(0.4743, 0.5047)	3137	93.47
1300 °C-8 h	0.91	(0.4746, 0.5048)	3134	93.58
1310 °C-8 h	0.99	(0.4746, 0.5044)	3132	93.45
1320 °C-8 h	0.63	(0.4748, 0.5045)	3130	93.54
1300 °C-4 h	0.40	(0.4743, 0.5046)	3137	93.44
1300 °C-12 h	0.75	(0.4746, 0.5047)	3133	93.55

## Data Availability

All data that support the findings of this study are included within the article.

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
