# Peer review of "Multifunctional Characteristics of BCTH:0.5% Sm3+ Ceramics Prepared via Hydrothermal Method and Powder Injection Molding"

_materials, 2023, doi:10.3390/ma16216910_

Round 1

Reviewer 1 Report

Comments and Suggestions for Authors

- The title is in accord with article

- The manuscript adheres to the journal's standards after revision

- This article contains new aspects, but the authors must underline the major findings of their work and explain how this study represents a progress to other similar published papers. Please provide comparison with other articles

- The Abstract section refers to the study findings, methodologies, discussion as well as conclusion. The Abstract section must be improved. The Abstract should refer to the study findings, methodologies, discussion as well as conclusion. In this form the abstract is too generally 

- The keywords permit found article in the current registers or indexes

- In the introduction it is not clearly described the state of the art of the investigated problem. More references are necessary. The references from last years are necessary for demonstrated that this study is actual

- The text can be understood by specialists from other domains

- In Tables are presented necessary results

- The literature is sufficiently critical, current, and internationally evaluated

- The size of the article is appropriate to the content

Reviewer 2 Report

Comments and Suggestions for Authors

Manuscript No: Materials - 2666626

 entitled: „Powder dependence of multifunctional characteristics of BCTH:0.5% Sm3+ ceramics prepared by hydrothermal method via powder injection molding

by Rong Zhuang , Qiushi Wang , Bijun Fang , Shuai Zhang , Xiaolong Lu  and Jianning Ding.

Manuscript refers results of structural, electrical, and magnetic studies of ferroelectric Sn-doped (BaCa)(TiHf)O3 ceramics, BCTH:S. Results obtained by XRD test, grain morphology, dielectric permittivity and dielectric loss coefficient temperature and frequency dependence,  and optical emission spectra features are reported analysed, and discussed. The Sn ions doping was performed to affect photoluminescence properties. Processing conditions were changed to tune these properties. The reference list of 55 papers seems sufficiently collected. The optical features are not commented in this review.

 #

Minor changes are recommended.

1/ The XRD analysis, showing correlation between the phase content and the processing conditions is convincing. Despite not all lines, hardly discerned, in the XRD pattern were indexed. This shortage should be commented in text.

2/ Electrical features presentation should be amended.

 It is better to use the term dielectric permittivity instead of dielectric constant, which is archaic. Anyway, the dielectric permittivity of BCTH:S markedly varies.

What was the temperature range of electrical measurements? It is not mentioned in the Experimental chapter. Why was it was measured for temperatures higher than 30 oC only?

3/ Fitting shown in Figure 5b is not perfect. It deviates from the claimed linear" dependence for the range close to the proposed phase transition temperature, where it would be more properly applied according to the phenomenological models of diffused phase transition (diffuseness coefficient ought to be estimated in a temperature range closer to the Tm) . Hence, the dielectric permittivity anomaly is not related to the normal, classical, ferroelectric phase transition. Consider the reasons of diffuse phase transition features and include such discussion to the manuscript.

4/ The grain size is in the micrometre range (1-4 micrometre) and in the sub-micro range (diameter > 100 nm). Hence, a size effect attributed to the nano-size effect is doubtful and exaggerated.

#

 In summary, I recommend minor revision of this manuscript. 

Reviewer 3 Report

Comments and Suggestions for Authors

Generally, the work is good and the results are interesting. Research methods selected by the authors are adequate to the subject of the paper.

The references are updated and consistent with the topic of work, 32 of 55 are published in the last decade (2013-2023).

Figure 3 is not very legible, please correct it or divide it into two separate Figures.

The topic of the paper is very interesting and worth publication.

I recommend the article for publication after a minor revision.
